# Quantification and Profiling of Early and Late Differentiation Stage T Cells in Mantle Cell Lymphoma Reveals Immunotherapeutic Targets in Subsets of Patients

**DOI:** 10.3390/cancers16132289

**Published:** 2024-06-21

**Authors:** Lavanya Lokhande, Daniel Nilsson, Joana de Matos Rodrigues, May Hassan, Lina M. Olsson, Paul-Theodor Pyl, Louella Vasquez, Anna Porwit, Anna Sandström Gerdtsson, Mats Jerkeman, Sara Ek

**Affiliations:** 1Department of Immunotechnology, Lund University, 221 00 Lund, Sweden; 2Department of Laboratory Medicine, National Bioinformatics Infrastructure Sweden, Science for Life Laboratory, Lund University, 221 00 Lund, Sweden; 3Division of Oncology and Pathology, Department of Clinical Sciences, Lund University, 221 00 Lund, Sweden

**Keywords:** mantle cell lymphoma (MCL), image analysis, deep learning, spatial omics, proteomics, tumor-immune microenvironment, CD57

## Abstract

**Simple Summary:**

In our study, we investigated immune regulation in a subtype of lymphoma called mantle cell lymphoma (MCL). Our goal was to understand how different types of immune cells, specifically T cells, behave in MCL. To do this, we combined image analysis and a technology called digital spatial profiling. We looked at tumor tissue from 102 MCL patients and analyzed different T-cell subsets. Interestingly, we found that late-stage differentiated T cells (CD57+) were more common in tumor-rich areas. These T cells also showed an increased expression of immune suppressive markers. We identified potential targets for treatment, such as CD47, IDO1, and CTLA-4. Additionally, we found that patients with sparse T-cell infiltration had an increased amount of GITR, which tentatively can be therapeutically targeted. Our findings shed light on previously unknown features of T-cell behavior in MCL.

**Abstract:**

With the aim to advance the understanding of immune regulation in MCL and to identify targetable T-cell subsets, we set out to combine image analysis and spatial omic technology focused on both early and late differentiation stages of T cells. MCL patient tissue (*n* = 102) was explored using image analysis and GeoMx spatial omics profiling of 69 proteins and 1812 mRNAs. Tumor cells, T helper (T_H_) cells and cytotoxic (T_C_) cells of early (CD57−) and late (CD57+) differentiation stage were analyzed. An image analysis workflow was developed based on fine-tuned Cellpose models for cell segmentation and classification. T_C_ and CD57+ subsets of T cells were enriched in tumor-rich compared to tumor-sparse regions. Tumor-sparse regions had a higher expression of several key immune suppressive proteins, tentatively controlling T-cell expansion in regions close to the tumor. We revealed that T cells in late differentiation stages (CD57+) are enriched among MCL infiltrating T cells and are predictive of an increased expression of immune suppressive markers. CD47, IDO1 and CTLA-4 were identified as potential targets for patients with T-cell-rich MCL TIME, while GITR might be a feasible target for MCL patients with sparse T-cell infiltration. In subgroups of patients with a high degree of CD57+ T_C_-cell infiltration, several immune checkpoint inhibitors, including TIGIT, PD-L1 and LAG3 were increased, emphasizing the immune-suppressive features of this highly differentiated T-cell subset not previously described in MCL.

## 1. Introduction

Mantle cell lymphoma (MCL), an aggressive subtype of B-cell lymphoma, has a complex tumor-immune microenvironment (TIME) that impacts tumor growth, progression and treatment outcome [1,2,3]. The composition of the TIME is highly variable across patients and includes various immune cell subtypes, such as T cells, macrophages, neutrophils and dendritic cells, but also soluble factors [4]. Exploring the variation in the cellular composition of the MCL-specific TIME and the associated functionality is pivotal for understanding the observed patient-to-patient heterogeneity, identifying novel targets and developing tools for patient stratification to different immunotherapeutic treatments [5]. T cells play a crucial role in governing the interplay between the immune system and cancer cells. Thus, an important part of tumor escape mechanisms includes the induction of senescence in T cells [6]. Reducing such immune-suppressive mechanisms is a key strategy for immunotherapy. Clinical and pre-clinical immunotherapy research has mainly focused on checkpoint inhibitors in MCL [7], including PD-L1/PD-1 [8,9], TIM-3 [10] and TIGIT [11]. In contrast to some other cancers, the clinical use of checkpoint inhibitors has so far been limited in MCL. To improve the applicability of immunotherapies in MCL, a deeper understanding of T-cell responses is warranted.

T-cell-associated antigens such as PD-L1 and FOXP3 have been shown to negatively impact MCL outcome [12], emphasizing the therapeutic potential for checkpoint inhibitors in subgroups of patients. Further, Nygren et al. showed that a high CD4:CD8 ratio in tissue was correlated to longer overall survival [12], justifying further investigations of T-cell subsets to optimize the stratification of patients in relation to immunotherapy. To achieve this, comprehensive studies that move beyond single plex biomarkers investigations are needed. We hypothesized that omics investigations of detailed T-cell subsets including CD8+ cytotoxic (T_C_) and CD4+ helper (T_H_) T cells at distinct differentiation stages, as defined by CD57 expression, may reveal biological insight and novel targets in MCL. CD57 is known to mark the late differentiation stage in T cells [13]. In solid tumors, CD57 has, in different studies, been associated with either senescent or proliferating T cells but the antigen has not been studied previously on T cells in MCL.

Recent advancements in deep learning-based image analysis (IA) have led to dramatic improvements in the identification, segmentation and classification of cells in tissue. Spatial omic and image analysis combined, provide access to comprehensive cell metrics and molecular information embedded in standard, diagnostic tissue specimens. In this study, we take advantage of such integrated workflows and apply spatially resolved transcriptomic (~1811 plex mRNA) and proteomic (~63 plex proteomic) analysis to a population-based cohort of 102 MCL patients. Specifically, we explore (i) the differences of T cells infiltrating tumor regions compared to when located adjacent to the tumor, (ii) the molecular profile of infiltrating CD57− and CD57+ subsets of T cells, which represent early and late differentiation stages, and (iii) the connection between the degree of T-cell infiltration (T-cell rich or T-cell sparse) in MCL and molecular features of tumor and T cells (Figure 1A). The study identifies possibilities for different immunotherapy strategies for patients based on the degree and type of T-cell infiltration, enabled by recent advances in spatial omics technology and machine learning strategies for image analysis and data integration.

## 2. Methods

### 2.1. Patient Cohort

Formalin-fixed paraffin-embedded (FFPE) tissue biopsies from MCL patient material (*n* = 102) belonging to the population-based Biobank of Lymphomas in Southern Sweden (BLISS, 2000–2014) were used in this study [14]. Clinicopathological information, including treatment, tissue origin, established risk factors and outcome, is available in Appendix A. Based on expert pathology review, tissue microarrays (TMA) were constructed with duplicate cores with a diameter of 1 mm from FFPE blocks and transferred into TMA receiver blocks via an automated device (ATA-27, Beecher Instruments, Sun Prairie, WI, USA). A total of 195 cores were arranged in three TMA blocks.

### 2.2. GeoMx™ Digital Spatial Profiling

#### 2.2.1. Multicolor Immunofluorescence (mIF) Staining

The first step of the spatial omics workflow includes the visualization of cells of interest in the tissue. Sections (4–7 μm thick) of the TMA were fixed on glass slides (Superfrost plus). Sections from each of the three TMA blocks were stained for mIF and scanned at 20X in the GeoMx instrument (Nanostring, Seattle, WA, USA) to collect (1) T-cell protein data, (2) MCL tumor protein data and (3) tumor B- and T-cell transcriptome data. Information on antibody clones is provided in Appendix A. All antibodies went through detailed optimization to determine the optimal concentration for the tissue used and staining was performed according to established protocols available from Nanostring (https://university.nanostring.com including MAN-10087-07, MAN-10089-06, and MAN-10116-04 for v2.2 SW all from February 2021). To visualize cells of interest, a tissue section was stained with (1) CD3, CD8, CD57 and nuclear counterstain Syto13, to identify four T-cell subtypes of interest: T_C,57−_ (CD57− CD3+ CD8+ early-stage differentiated T cytotoxic cells), T_H,57−_ (CD57− CD3+ CD8− early-stage differentiated T helper cells), T_C,57+_ (CD57+ CD3+ CD8+ late-stage differentiated T cytotoxic cells) and T_H,57+_ (CD57+ CD3+ CD8− late-stage differentiated T helper cells) (Figure 1B, middle panel). The presence of double negative T cells (CD4− and CD8−) is rare (<3%), justifying the definition of T helper cells by the absence of CD8 on CD3+ cells [15]. A second tissue section was stained with (2) the tumor/B-cell markers CD20, CD3 and Syto13 to allow the visualization of two spatially defined regions, the tumor-rich and tumor-sparse regions where transcripts/proteins were quantified. From this section, protein data were collected from CD20+ cells (Figure 1). For spatial transcriptomics, a third tissue section was stained with (3) CD20, CD3 and CD8 (Appendix A) to identify malignant CD20+ tumor cells, T_C_ (cytotoxic T cells, CD3+ CD8+) and T_H_ (helper T cells, CD3+ CD8−).

#### 2.2.2. Region of Interest (ROI) Selection

The selection of the ROI was visually guided by the CD20+ and/or CD3+ staining to collect data in either tumor-rich regions dominated by CD20+ (from now on referred to as tumor-rich regions, *n* = 102) or regions abundant in CD3+ T cells but with few CD20+ cells (from now on referred to as tumor-sparse regions). In general, tumor-rich areas are dominated (>80%) by MCL cells, while tumor-sparse areas are dominated (>80%) by non-MCL cells such as T cells, other immune cells and structural cells. For the T-cell subset analysis (where no CD20 morphology staining was available) ROIs were placed in the same part of the tissue as the adjacent slide with CD20 staining, and additionally guided by the CD3-rich regions. Less than half of the MCL tissue biopsy cores had tumor-sparse regions, and proteomic data were collected from 39 patients and transcriptomic data from 31 patients from such tumor-sparse regions (Figure 1B, Appendix A). Paired data with both T_H_ and T_C_ data from tumor-sparse regions were available from 25 patients and were used for comparisons of differentially regulated transcripts/proteins between the two regions. Example images of patient tissue cores containing both regions are shown in Appendix A. All stainings and annotation of regions were subjected to careful pathology review.

Within the ROIs, cell-specific omics data were collected from cell-types of interest. These sub-regions within a selected ROI are called areas of illumination (AOIs). The summary of the number of ROIs/AOIs collected for the cohort is provided in Appendix A. The number of AOIs collected per patient is presented in Appendix A. Overall, proteomic data were collected in 482 ROIs, including data from tumor cells (239 AOIs) and T-cell subsets (588 AOIs). For transcriptomics, data were collected in 327 ROIs including tumor (159 AOIs) and T-cell subsets (252 AOIs).

Not all T-cell subtypes were sampled in each ROI, either due to the absence or low number of that subset (the limit of detection per AOI is set at ≥20 cells for proteomic and ≥100 cells for transcriptomic data), or due to the inaccurate segmentation of cells caused by varying local background in the fluorescence image. A summary of the collected AOIs and clinicopathological information are provided in Appendix A.

#### 2.2.3. Retrieval of Probes for Proteomic and Transcriptional Analyses

The collection and quantification of the protein or mRNA targets in each AOI was performed according to protocols provided by Nanostring. Briefly, oligos conjugated to bound reagents (63 antibodies or 1811 mRNA probes with respective controls) were cleaved off by ultraviolet light and collected into individual wells on a microtiter plate for the separate profiling of each cell subset, in each AOI. See Appendix A for information and the abbreviation of the targeted panel for proteome and Cancer Transcriptome Atlas (CTA) assays.

#### 2.2.4. Pre-Processing of GeoMx™ Data

The data were scaled by the respective AOI collection area and normalized using cyclic loess to account for differences in cell sizes and AOI areas. For mRNA analysis, transcripts were filtered and removed if >25% of the signal-to-noise ratio (SNR) across patients was <1.05 in all three cell types, leaving 1482 transcripts for biological explorations. For both transcriptomic and proteomic data, the normalization process was done separately for the T-cell subset AOIs and CD20 AOI. The number of AOIs per patient and nuclei count per cell type in the proteomic and transcriptional analyses are shown in Appendix A.

### 2.3. Image Analysis, including Cell Segmentation and Classification

Image analysis (IA) was performed using Cellpose, a deep learning-based UNET architecture [16], using images collected from the T-cell proteome section (Figure 1B). Segmentation and classification were performed both for tumor-rich and tumor-sparse regions, separately. Most tissue was of high quality and the full tissue area could be used. Folded tissue and regions with high auto-fluorescence were excluded from the analysis. The segmentation of cells was performed by using transfer learning and the fine-tuning of the pre-existing Cellpose models using the Cellpose graphical user interface (GUI) [17]. This allowed the extraction of updated weights upon which a final model for our dataset could be based, see Figure 2. To retrieve information on the total number of nucleated cells, the Cellpose ‘nuclei’ (v2.2.2) [16,17] model was fine-tuned based on Syto13 (Figure 1C and Figure 2A,B) using 42 randomly isolated images (size −256 × 256 px or 128 × 128 px, depending on the tissue quality). For phenotype-based cell segmentation and classification, the Cellpose ‘cyto’ model was finetuned on a single channel using 121 image crops (256 × 256 px) on either CD3, CD8 or CD20.

To classify cells, the segmentation masks were overlapped using the centroid of the nuclei segmentation mask and based on the presence or absence of each marker, classified into the four cell types (Figure 1C and Figure 2A). Classification was done in QuPath (v.0.4.4) with the Cellpose extension (v.0.8.0) (https://github.com/BIOP/qupath-extension-cellpose (accessed on 12 June 2024)) [16,17,18]. IA was performed for the full tumor-rich region, and cores were trimmed to remove tumor-sparse regions or areas where the segmentation had failed post inspection. IA was also performed on ROIs in tumor-sparse regions (Appendix A). Quality control was performed using visual inspection and when segmentation was incorrect (3.2% of cores), manual counting was applied instead (Appendix A). Cell frequencies are defined as the cell counts over the total number of nucleated cells. When comparing the relative abundance of T-cell subsets, such as T_H_/T_C_, ratios, or sometimes percentages, are used.

### 2.4. Statistical Analysis

All data analysis was performed using R (v. 4.1.0) and R studio (v. 1.2.5001, R Foundation for Statistical Computing, Vienna, Austria). Statistical significance was considered for comparisons resulting in a *p*/q-value < 0.05. Data visualization was done using R package “ggplot (3.5.1)”, except heatmaps/heat correlation plots which were made using the packages “ComplexHeatmap (2.16.0)” and “heatmaply (1.5.0)”.

Shannon’s diversity index (SDI) uses relative abundance to assess the complexity of a population. The higher the index, the greater the complexity and diversity with more sub-populations contributing to the total population [19,20,21]. SDI was applied to image data from the T-cell subsets to describe variation among the four T-cell subsets and summarize it into a single score.

For the quantitative analysis of the cell classification models, intersection over union (IoU) was calculated against manual annotation (ground truth), using a subset of 11 cropped images. With a 0.5 IoU threshold, precision, recall (sensitivity), accuracy and F1 score were calculated for comparison.

Linear mixed models (LMM) were fitted using R package “lmerTest (3.1–3)” to account for repeated sampling per patient. In these models, patient ID was set as the random effect. If the fixed effect was continuous, it was scaled and centered around zero prior to performing LMM analysis. To account for multivariate analysis, *p*-values extracted from LMM analysis were adjusted for false discovery rate (FDR) using the Benjamini–Hochberg method and the FDR cut-off was set at 0.05.

For correlation analysis, Spearman correlation was used. For all two-group comparisons, the Shapiro–Wilk test was used primarily to evaluate normality in each distribution along with quantile–quantile plots and histograms. For two-group comparisons, either the *T*-test or Wilcoxon test was used, depending on the result for the Shapiro–Wilk test.

For multigroup comparison, multivariate ANOVA was performed followed by Tukey’s Honestly Significant Difference (HSD) test to identify analytes of interest differentiating T-cell subsets.

Gene set enrichment analysis (GSEA) was performed using R package “clusterprofiler (4.8.3)” using gseKEGG function. Deconvolution was performed using the SpatialDecon workflow by Nanostring [22] where mRNA (T_H_, T_C_ and CD20) data were normalized together to enable relative comparison between the three cell types.

To compare clinico-pathological features with cell type abundance, chi-square analysis was performed. If the number of occurrences was less than five, the Fischer exact test was used to confirm the results. The univariate Cox proportionality hazard model and Kaplan–Meier analysis, along with log-rank statistics, were done using the R packages “survival (3.5–5)” and “survminer (0.4.9)”. Survival analysis was done using overall survival (OS), defined as the time in years from diagnosis to current accessible follow-up (until 2022). To assess whether Cox regression was applicable, a test for Schoenfeld residuals was performed prior to regression. To dichotomize continuous variables to categorical, maximally rank statistics (survminer: survcutpoint) against overall survival was used with a minimum group distribution set at 0.1.

To develop integrated models combining mRNA and protein data, “Data Integration Analysis for Biomarker discovery using Latent variable approaches for Omics studies” (DIABLO) was applied using package “mixOmics (6.24.0)” [23,24]. Data integration could be performed on tissue from 62 patients, where joint datasets for proteomic (T_H,57−_, T_C,57−_, CD20) and transcriptional profiles (CD20 and Tc) were available. The transcriptional profiles from T_H_ cells were not included as it would limit the number of patients with complete datasets required for DIABLO-based integration. The DIABLO model was trained using a design matrix of 0.1 between the datasets and 1.0 between the target variables, to allow for maximum discrimination. A 5-fold repeated cross validation with 100 repeats was used to determine the optimal number of principal components, which was used to tune the model using the perf function. To determine the optimal analytes to retain per dataset, the tune.block.splsda function was used with a 5-fold cross validation repeated 50 times. The Mahalanobis/Centroid distance was selected as the distance metric based on the results of cross validation, for prediction. For the mRNA data, the number of tested analytes was 10–50 with a gap of 10 while for the protein data, it was 5–15 with a gap of 5. Finally, a permutation test where the labels were randomly shuffled with 1000 permutations was used to validate the model performance against randomness [25]. Each dataset was evaluated by analyzing the selected analyte’s contribution (loading) for the principal components created by DIABLO.

## 3. Results

In this study, a combination of image analysis and spatially resolved multi-omics was applied to determine frequencies and molecular profiles of tumor cells, along with early- and late-stage differentiated T-helper and T-cytotoxic subsets (T_H,57−_, T_C,57−_, T_H,57+_, T_C,57+_) in a population-based cohort of 102 MCL patients (Figure 1).

### 3.1. Image Analysis to Retrieve Information on T-Cell Subset Composition in MCL TIME

#### Fine-Tuning of Deep Learning-Based Image Analysis Models Is Required to Retrieve Accurate Measurements of Cell Frequencies in MCL

Cell frequencies are important metrics to describe the biology of the MCL TIME. mIF (DNA, CD3, CD8 and CD57) was used to determine the abundance and diversity of the four subsets of T cells in 102 MCL patients in tumor-rich and tumor-sparse regions, respectively (Figure 2A, Appendix A). The fine-tuned Cellpose model provided accurate nuclei identification and cell segmentation/identification and was visually compared to untuned models (Figure 2B,C). Visual comparison to a random forest (RF)-based cell classifier showed that the applied workflow provided better accuracy in cell identification, particularly in cases with variable staining background (Figure 2C). Overall, this optimized workflow handled varying backgrounds, staining artifacts and differences in staining intensity in most tissue cores (Figure 2C,D), and accurately identified and classified cells (Figure 2E).

### 3.2. A proportion of Infiltrating T Cells in MCL Are CD57+

In tumor-rich regions, total CD3 frequency varied from 1.7% to 16.8%. The median (±SD) frequencies of the four T-cell subsets investigated were 4.9 ± 4.6% (T_H,57−_), 2.5 ± 2.9% (T_C,57−_), 0.5 ± 0.4% (T_H,57+_) and 0.7 ± 1.5% (T_C,57+_) (Figure 3A). Thus, T cells in tumor-rich regions were dominated by T_H,57−_ (56.6 ± 16.4% out of all CD3+ cells), and T_C,57−_ (28.7 ± 10.6%) while the CD57 positive subsets constituted 4.6 ± 3.1% and 8.3 ± 8.9%, respectively (T_H,57+_ and T_C,57+_) (Appendix A). The frequency of T-cell infiltration in the tumor-rich regions was highly variable between patients (Appendix A). Of interest, no correlation was observed between the T_57+_/T_57−_ ratio and CD3+ cell frequency (R: −0.094, p~ns) emphasizing that the highly differentiated subsets frequency is less tightly connected to total T-cell infiltration compared to less differentiated subsets (Appendix A).

### 3.3. T_C_ and Late-Stage Differentiated, CD57+ T-Cell Subsets Are Enriched among MCL-Infiltrating T Cells

Differences in T-cell subtypes between tumor-rich and tumor-sparse regions were compared in a subset of patients (*n* = 39) where both regions were available (Figure 3B and Appendix A). The ratio of T_H_/T_C_ (CD4/CD8) was lower in tumor-rich regions, suggesting a higher percentage of T_C_ among tumor-infiltrating T cells in MCL. In paired analysis, the percentage of T_C_ vs. T_H_ (irrespective of CD57 status) was 34.7% in tumor-rich vs. 25.3% in tumor-sparse regions (Appendix A). Regarding cell differentiation stage, the T57+/T57− cell percentage (16.5 vs. 4.2%) was significantly higher in the tumor-rich compared to tumor-sparse regions (Appendix A, Figure 3B). Thus, T_C_ and CD57+ T-cell subsets are highly enriched among infiltrating T cells in tumor-rich compared to tumor-sparse regions.

### 3.4. Diversity among T-Cell Subtypes Is Not Associated with Total CD3 Infiltration

The Shannon-Wiener Diversity Index (SDI), allowed us to explore the complexity of the T-cell populations, including the investigated four subsets. A low SDI score indicates that one of the T-cell subsets is dominating among the total T cells, while a high score indicates that multiple T-cell subtypes contribute to the total number of T cells. We show that T-cell subset complexity (SDI score) was independent of total CD3 frequency (Figure 3C,D) but associated with the presence of late T-cell differentiation stages (Figure 3E). Thus, increased T-cell infiltration per se is not associated with the presence of well-differentiated T-cell subtypes.

### 3.5. T_C,57+_ Cells Are Associated with Highly Proliferative MCL, while Total CD3+ T-Cell Infiltration Is Positively Associated with Favourable Prognosis

We investigated whether the abundance of the four T-cell subsets in tumor-rich regions was associated with established clinicopathological parameters, including gender, age, *TP53* mutational status, Ki-67 (IHC) and p53 (IHC) protein levels (Appendix A). Female gender was shown to be positively correlated to high T_c_ cell frequency (median cut-off, chi-square *p* = 0.05). High Ki-67 expression in MCL cells (>30%) was shown to be positively correlated with high T_C,57+_ frequency (median cut-off, chi-square *p* = 0.0051), and group-wise comparison supported the trend but was not significant (Appendix A). Survival analysis showed that a higher infiltration of total T cells and individual T-cell subsets T_C,57−_ and T_H,57+_ were associated with favorable prognosis in unadjusted models (Appendix A, Appendix A). The optimal cut-off, (8.4%) for CD3 was defined using maximally ranked statistics and was in line with our previous study cohort (10.1%) [26], but prognostic information on CD57+ T cells has not previously been reported in MCL. We further compared the median overall survival for patients where either both tumor-sparse and tumor-rich regions (5.69 years), or only tumor-rich regions (4.22 years) were available, but the difference was not significant using log-rank statistics. We conclude that the enrichment of infiltrating T_C_ and CD57+ subsets indicates that they mediate important functions in the MCL TIME. The tentative functionality of the individual subsets was further investigated using molecular profiling as described below.

### 3.6. Molecular Comparison of T_H_ and T_C_ Subtypes in Tumor-Rich Versus Tumor-Sparse Regions of MCL Tissue

The molecular profile of T_H_ and T_C_ cells was compared in tumor-rich versus tumor-sparse regions. Targeted proteomic (*n* = 63) (Appendix A) and transcriptomic (*n* = 1482) panels were explored (Appendix A). Of note, too few AOIs from CD57+ T cells were available to allow an analysis of such subsets in tumor-sparse regions, and analysis was focused on CD57− T_H_ and T_C_ subsets.

### 3.7. Deconvolution Analysis Supports Data on Well Differentiated T Cells of Memory Type among Infiltrating T Cells in MCL

Cells in the dense MCL tissue are in direct cell-to-cell contact, leading to AOIs also including information from the most adjacent cells, beyond the targeted cell type. Despite being a technical drawback, this can be explored biologically. Considering this, the difference in cell type abundance between tumor-rich and tumor-sparse regions was assessed using the deconvolution of the transcriptome data from 25 patients and subsets of structural cells (Figure 4A), different immune cells (Figure 4B), T cells (Figure 4C) and B cells (Figure 4D) were investigated. As expected, this showed that macrophages, myeloid dendritic cells and plasmacytoid dendritic cells were more abundant in tumor-sparse compared to tumor-rich regions, indicating close contact between T cells and antigen presenting cells (APC) in the tumor-sparse region. Furthermore, regulatory T cells (T_regs)_ were predicted to be of higher proportion in the Tc compartments in tumor-sparse vs. tumor-rich regions. In line with previous knowledge, T_regs_ were predicted to be more common among T_H_ compared to T_c_ in tumor-rich regions. However, it should be highlighted that no difference in T_regs_ proportion was seen comparing T_H_ and T_C_ in tumor-sparse regions. In addition, CD4+ T-cell memory cells were predicted to be more abundant in tumor-sparse regions while CD8+ T-cell memory cells were more abundant in tumor-rich regions. This suggests that CD8+ T cells not only are more abundant among infiltrating cells as reported above, but also more differentiated, which is in line with the reported CD57+ enrichment.

### 3.8. T Cells in Tumor-Sparse Regions Show Increased Use of TNF-Related Pathways in T_H_ Cells and Higher Levels of T-Cell Suppressive Proteins such as VISTA, TIM3, LAG3, and IDO1

Gene set enrichment analysis identified common and unique pathways activated in T_C_ and T_H_ collected in tumor-sparse regions, when compared to tumor-rich regions (Figure 5A). Of note, TNF-related signaling in the Tc and PD-L1/PD-1 checkpoint pathway was increased in T_H_ collected in tumor-sparse regions. Transcripts associated with the pathways were validated by differential gene expression analysis (Figure 5B,C, Appendix A). For the TNF signaling pathway, 15/23 transcripts were confirmed to be individually de-regulated when comparing the expression in tumor-rich versus tumor-sparse regions. These included *MAPK3K5*, *MAPK14*, *BIRC3*, *CASP8*, *TNFRSF1B*, *AKT3*, *CASP10*, *IL18R1*, *CYLD*, *PIK3R1*, *TNFRSF1A*, *TRAF1*, *FOS*, *SOCS3* and *CCL2*. Key transcripts include *CCL2*, which is involved in leukocyte recruitment, regulators of PI3K-AKT signaling (*TRAF1*, *PIK3R1* and *AKT3*) and MAPK signaling (*MAPK14*, *MAP3K5*, *CYLD*, *TNFRSF1A/B* and *CASP8/10*). As expected, signals to attract and retain T cells were enriched in these tumor-sparse regions, which are abundant in T cells. These included, for example, *CCL21*, *CCL19*, *SELL* (*selectin L*) and *CCR7* (Figure 5B,C, Appendix A), which are all mediators of T-cell/APC recruitment and T-cell adhesion [27,28,29,30].

Differential protein expressions in tumor-sparse compared to tumor-rich regions reveal both common and uniquely regulated proteins in T_H,57−_ and T_C,57−_ cells (Figure 5D,F and Appendix A). For example, CD45RO was enriched in T_H,57−_ in tumor-sparse regions, which is in line with the predicted higher abundance of memory TH cells in tumor-sparse regions, based on deconvolution results in Figure 4. Of note, the T-cell suppressive proteins IDO1, VISTA and TIM-3 also had higher expression in tumor-sparse regions. T-cell subsets in the tumor-sparse region also showed higher relative abundance of fibronectin, SMA and CD34, which are markers of neovascularization and angiogenesis. Deconvolution predicted an increased abundance of fibroblasts in the tumor-sparse region, and the proteins may also derive from such cells. Additionally, we observe an increased expression of STING in tumor-sparse regions, which is known to increase IDO1 activity [31].

Other proteins were only differentially regulated in either T_H,57−_ or T_C,57−_ cells. These included LAG3, which was significantly higher on T_H,57−_ cells in tumor-sparse regions. LAG3 is an inhibitory receptor known to be highly expressed on exhausted T cells [32]. ICOS, which is a hallmark of CD8+ tissue-resident memory T cells [33], had higher expression in T_C,57−_ in tumor-sparse regions. Thus, CD57− subsets in the tumor-sparse region showed markers known to be associated with exhaustion together with the expression of the protein memory marker CD45RO. A high expression of CD163 in tumor-sparse regions suggested the potential presence of CD163+ M2 type macrophages/monocytes, which supports the results from the deconvolution analysis.

For the tumor-rich regions, gene set enrichment analysis showed that T cells were in close contact with tumor cells, with an upregulation of pathways associated with tumorigenesis including B-cell receptor signaling and DNA replication (Figure 5A).

In summary, a comparison of tumor-sparse and tumor-rich regions showed that memory CD8+ T cells were predicted to be more abundant among infiltrating T cells. Furthermore, the infiltrating T_C_ had a higher expression of transcripts involved in TNF signaling compared to tumor-adjacent T cells in tumor-sparse regions, indicating that they are activated. Based on the fact that several tumor suppressive proteins (IDO1, VISTA, TIM-3, LAG-3 and ICOS) had a higher expression in the tumor-sparse region, we conclude that a wide range of T-cell suppressive features may hamper the expansion of T cells in regions also adjacent to the tumor.

### 3.9. Infiltrating T Cells in Tumor-Rich Regions: Identification of Unique Analytes among Early and Late T_H_ and T_C_ Subsets in MCL

The molecular features of infiltrating (sampled in the tumor-rich region) early and late T_H_ and T_C_ subsets have not been studied in MCL and such information can provide insight in lymphoma-related immune dysregulation. A transcriptional analysis of infiltrating T_H_ and T_C_ cell subsets was performed along with the proteomic profiling of four T-cell subsets, T_C,57−_, T_H,57−_, T_C,57+_ and T_H,57+_.

#### Transcriptional Analysis Reveals Co-Regulation of T_regs_ Markers in T_H_ Cells and Expression of Antigens Associated to Late Differentiation in T_C_ Cells

The expression of 58 transcripts was associated with T_H_ cells and the expression of 62 transcripts was associated with T_C_ cells (Appendix A, Appendix A). In T_H_ cells, co-regulation was observed between *IL7R, LEF1*, *TCF7*, *IL6ST*, *CD4* and *TECR* (Figure 6A). *TCF7* and *LEF1* are known co-regulators for various signaling mechanisms including the maintenance of CD4 T-cell receptor expression [34], the activation of CD4 T cells [35] and immune suppressive functions by T_regs_ [36]. In addition, a higher expression of *FOXP3*, *CTLA4* and *IL6R*, which are all important for T_regs_ function [37], was observed in T_H_ vs. T_C_, as expected (Appendix A).

In T_C,_ differentially expressed transcripts that were co-regulated included *CD8A*, *CD8B*, *GZMA*, *GZMK*, *GZMH*, *PRF1*, *LAG3*, *NKG7*, *TNFRSF9*, *CCL4*, *CCL5*, *EOMES*, *CCR5*, *CTSW* and *FASLG* (Figure 6A)_._ Chemokines *CCL5* and *CCL4* have been associated with the modulation of TIME, particularly with respect to immune cell infiltration and tumor progression [38]. In addition, *SLAMF7*, *TARP*, *TRGC1.2*, *CD2*, *SH2D1A* and *CCL3.L1*, *KLRG1*, *KLRK1*, *XCL1.2* and *IL2RB* (*CD122*) had higher expression in T_C_ than T_H_ (Appendix A). *XCL1* is a chemokine mainly produced by activated T_C_ or NK cells and has been correlated to PD-L1 expression on tumor cells [39]. *EOMES* has been shown to differentiate between CD57+ cells with low cytotoxicity but proliferative capacity and *IL7R* expression (EOMES^high^), and CD57+ cells with pronounced cytotoxic functions but more terminally differentiated phenotype (EOMES^int^) [40]. Several of the markers are consistent with a late-stage differentiated CD8 effector cell phenotype, including *KLRG1* [41], in line with the observed increased frequency of T_C,57+_ among infiltrating T cells as reported above.

A proteomic analysis of T_C,57−_, T_H,57−_, T_C,57+_ and T_H,57+_ subsets to identify unique expression profiles shows that inhibitory and stimulatory receptors are co-expressed and that CD57+ is associated with LAG3, 4-1BB, PD-L1 and PD-L2 expression.

Unique proteomic characteristics of each of the T-cell subsets, T_C,57−_, T_H,57−_, T_C,57+_ and T_H,57+_, were determined using multi group comparisons. In total, 26 differentially expressed proteins were identified (Figure 6B and Appendix A). The PCA biplot shows the contribution of the 26 proteins in segregating the four T-cell subtypes, particularly the overall T_C_ and T_H_ subsets (Figure 6C).

Early (CD57−) and late (CD57+) differentiation stages of T_C_ and T_H_ subtypes were compared (Figure 6B, right panel). CD27, PD-L1 and 4–1BB were more abundant in both T_C,57+_ and T_H,57+_ compared to CD57− subsets, indicating that these cell types both express markers of T-cell expansion (4–1BB, CD27) and inhibitory molecules (PD-L1) at the same time (Figure 6B,D). T cells with dual expression of both co- and inhibitory molecules have previously been described as polyfunctional, although the exact role of such cells is unclear [42].

In T_C,57+_, uniquely differentially regulated proteins compared to the other T-cell subsets included a relatively higher expression of, among others, LAG3, ICOS and CD68. The expression of LAG3 indicates T-cell exhaustion [43] and ICOS has been associated with IL-10 production and CD8+ tissue-resident memory T cells [33]. The relatively increased levels of CD68 indicate that macrophages are proximal to T_C,57+_. T_C,57−_ had a uniquely higher expression of, among others, GITR.

T_H,57+_ cells had higher levels of PD-L2, GZMA, GITR and CD45RO, while the CD57− subset had higher levels of FOXP3, CD25 and ARG1 (Figure 6B, right panel). The expression of PD-L2 is in line with the definition of CD57+ T cells as suppressive T-cell subsets. The increased expression of FOXP3 and CD25 in T_H,57−_ indicate that T_regs_ mainly are of low differentiation stage and lack CD57 expression. The multi-group differential expression analysis was validated using pair-wise comparisons, which also identified additional regulated proteins comparing early and late differentiation stage, such as VISTA, which was enriched in T_H,57−_ compared to T_H,57+_. TIM-3 was enriched in T_C,57+_ compared to T_C,57−_ (Appendix A).

The expression levels of PD-1, PD-L1 and PD-L2 were visualized separately for the four subsets of T cells (Figure 6D). An increased expression of PD-L1 and PD-L2 was associated with the CD57+ subtypes of both T_H_ and T_C_ and confirmed the immune suppressive features of these late differentiated subtypes. No difference in PD-1 was detected, in contrast to a few previous reports on other cancers [44,45].

Furthermore, differences between cytotoxic and helper T cells were determined. Both early-stage and late-stage differentiated T_H_ cells were compared to their T_C_ counterparts, respectively. The comparison mostly highlighted expected differences such as higher CD4 and lower CD8, GZMA and GZMB in both T_H,57−_ and T_H,57+_ compared to the respective T_C_ counterpart (Figure 6B, Appendix A). Uniquely differentially expressed proteins in T_H,57−_ compared to T_C,57−_ include higher levels of CD25, FOXP3, ICOS, CD45 and STING [46] and lower levels of 4–1BB (Figure 6B, left panel). Differentially expressed proteins in T_H,57+_ compared to T_C,57+_ include higher levels of GITR (Figure 6B right panel, Appendix A). *GITR* is a co-stimulatory surface receptor on T cells and shapes humoral immunity by controlling the balance between follicular T helper cells and regulatory T follicular cells [47]. In T_C,57+_, LAG3 and OX40L were more abundant. LAG3, as previously described, is a known check-point inhibitor and normally expressed by activated T cells [48], while OX40 signaling enhances primary and memory T-cell response and promotes antitumor activity [49]. Thus, we conclude that LAG3 is higher on T_H_ in tumor-sparse compared to tumor-rich regions as reported above, but that the relative expression is highest on T_C,57+_ cells in tumor-rich regions (Appendix A).

We conclude that both markers of expansion and inhibitory molecules, such as PD-L1, PD-L2, ICOS and LAG3, were positively associated with the late differentiated CD57+ T-cell subtypes compared to CD57− T cells.

### 3.10. CD47 Don’t Eat Me Signals Are Associated with High Total CD3 while CXCL9 Is Associated with High T_C,57+_ Frequency

The intra-patient variation in measured CD3+ T-cell frequency was assessed and found to be acceptable, and aggregated mean values were used in subsequent analysis (Appendix A). T-cell frequency was positively associated with outcome as shown above (Appendix A). Data integration was performed to identify transcripts/proteins that may explain the connection between outcome and T-cell frequency. The analysis identifies differentially expressed transcripts and proteins in both tumor cells and T-cell subsets that are predictive of high (>8.4%) or low (<8.4%) T-cell infiltration.

#### 3.10.1. High Level of Total CD3+ T-Cell Infiltration Is Associated with Increased Levels of CD47, IL7R and Key Components of Antigen Presentation

DIABLO (permutation test, *p* < 0.005) identified 15 transcripts in the CD20+ tumor cells (AOIs) to be positively associated with T-cell-rich MCLs (Figure 7, Appendix A), including *IL7R*, *CD47*, *CD80*, *CD84* and *NLRC5*. *IL7R* is associated with the attraction of T cells to the TIME. CD47 is an important inhibitory molecule involved in the evasion of immune clearance by macrophages that carry the ligand SIRPα [50]. CD80 is a co-stimulatory molecule essential for T-cell activation. CD84, which is a self-binding immunoreceptor belonging to the signaling lymphocyte activation molecule (SLAM) family, is determinative of high T-cell infiltration and activity. Lastly, *NLRC5*/*CITA* induces the expression of genes encoding critical components of the MHC class I pathway, which is essential for the cancer antigen presentation and recruitment/activation of cytotoxic T cells [51]. An increased expression of ICOS indicates that T cells are involved in antigen recognition, but the function can be dual with either anti-tumor or tumor promotion responses [52]. In CD20+ AOIs, 15 transcripts were negatively associated with T-cell-rich MCL and included genes related to cell cycling, DNA repair and mobility, all associated with aggressive disease. For example, this included *TUBB*, *SPOP*, *LAMA3*, *CDC20*, *BIRC5*, *H2AX* and the transcription factor *FOXM1* (Figure 7, Appendix A).

In addition to the analysis of CD20+ tumor cells, T cells were also investigated to increase our understanding of how the infiltrating T cells differed on the molecular level in tumors with T-cell-rich or T-cell-sparse TIMEs. As expected, few analytes differed, as identical cell types were compared only based on differences in the MCL TIME composition. Results showed that several markers associated with CD8+ T-cell response were positively associated with T-cell-rich TIME including IDO1 (both T_H_ and T_C_), CD45RO, *ECSIT* and co-stimulatory molecule *JAML* in T_C_ [53]. A few T_C_ transcripts were associated with T-cell-sparse MCL and included transcripts associated with transcription (*GTF3C1*) and alternative splicing (*MBNL3*) (Figure 7, Appendix A). The protein GITR, a marker also associated with T_regs_ was also higher in regions of low T-cell infiltration. Studies have shown that reverse signaling via GITR can induce IDO1 expression [54]. Correlation analysis reveals an inverse relationship between IDO1 and GITR in MCL (Appendix A).

#### 3.10.2. Differences in T_C,57+_ Infiltration Are Associated with Increased Proliferation, and Secretion of CXCL9 in Tumor Cells and Expression of CD45RO, TIGIT, PD-L1 and 4–1BB in T_C_ Cells

As reported above, infiltration of T_C,57+_ cells vary across MCL patients with a positive association to proliferation index (see Appendix A). To understand the molecular mechanisms related to increased abundance of late differentiation stage Tc infiltration, DIABLO (permutation test < 0.007) was used to define transcripts and proteins associated with and predicted T_C,57+_ infiltration above or below the median value (0.69%, Appendix A). Focusing on the CD20+ tumor cells, the transcriptomic analysis showed that *CXCL9* was most important for the classification of patients with either high or low T_C,57+_ infiltration. *CXCL9* is important for recruitment of T cells and has been shown to correlate with anti-tumor activity [55]. Other differentially regulated transcripts were involved in cell-cycling, DNA replication and repair such as *PARP9*, *MCM2* and *STAT1* (Figure 8, Appendix A). In the T cells, proteomic analysis showed that, like total CD3+ T cells, IDO1 was associated with high infiltration of T_C,57+._ Most relevant differentially regulated transcripts/proteins in T-cell subsets reflected the higher content of cytotoxic T cells such as *PRF1*, *LYZ*, *EOMES*, *CCR5* and *NKG7* (Figure 8, Appendix A). Of interest, the immune-check point inhibitors *TIGIT* and PD-L1 were positively associated with high T_C,57+_ frequency. In line with the well differentiated stage of an increased promotion of T cells, CD45RO, 4–1BB and the transcription factor *PRDM1* was positively associated with high T_C,57+_ frequency.

In summary, a combined analysis of the level of T-cell infiltration and associated molecular profiles in tumor cells showed that the *CD47* immune evasion marker was upregulated by tumor cells in T-cell-rich TIMEs, while *CXCL9* was specifically associated with a high degree of T_c_ of late differentiation stage.

The summary of all of the main findings and results of this study is provided in Figure 9.

## 4. Discussion

T-cell-directed immunotherapies have revolutionized cancer care, but patients show heterogenous response to treatment. This diversity warrants improved understanding of the TIME and the connection between the infiltration, spatial localization and functionality of different T-cell subsets in cancer. It has been shown that subtypes of B-cell lymphomas have distinct types of TIMEs, ranging from Burkitt´s lymphoma that largely lacks immune infiltration to Hodgkin´s lymphoma that is dominated by immune infiltration and with few tumor cells [56]. In this context, MCL TIME has been classified as dominated by immune-escape or re-education mechanisms. However, until now, no study has characterized subsets of immune cells in relation to spatial proximity to tumor cells and the degree of infiltration of T cells of early and late differentiation stages.

Here we set out to characterize T cells in the MCL TIME, focusing on the impact of CD4 and CD8 T-cell differentiation, as defined by CD57 expression. CD57 is considered a marker of differentiation and early studies suggested that CD57+ T cells were terminally differentiated and senescent [13]. More recent studies show that CD57−expressing cells can be proliferative and in a hybrid differentiation stage, defined by co-expression with early differentiation markers such as CD28 and CD27. These cells thus show functional features of both early effector and end-stage differentiation senescence [57]. A TIME characterized by both an impaired differentiation of CD8+ cells and domination by cells lacking CD57 is associated with poor prognosis in patients carrying such tumors [12]. In contrast, fully differentiated CD57+ T-follicular helper cells correlate with poor prognosis in follicular lymphoma [58] and with advanced disease in CLL [59].

To retrieve information on cell frequencies and spatial localization, we took advantage of the recent development of deep learning strategies in image analysis [60] that allow the collection of cell metrics in large patient cohorts. Comparison with alternative strategies showed superiority in the accuracy of the segmentation and classification of cells, leading to trustworthy estimations of rare T-cell subsets. The reliability of Cellpose segmentation has previously been shown for diverse sets of solid tumors using CODEX datasets [61] but has, to our knowledge, not been applied to lymphoma-derived patient-tissue images or combined with spatial omics data to explore biological diversity in TIME related to tissue composition.

Using image information extracted by a fine-tuned Cellpose model, we show that T_H_ dominates in numbers, but that T_C_ is enriched among infiltrating T cells in tumor-rich regions compared to T cells in tumor-sparse regions. Thus, we conclude that there is an active infiltration of cytotoxic T cells in MCL TIME. The frequency and ratios of T cells in late differentiation stage have not been reported before, but T_H_ and T_C_ frequencies were in line with previous studies using IHC by ourselves [26] and Vasallo et al. [62].

CD57 positivity in T_H_ and T_C_ was rare (lower than 1% of total cells) but constituted a substantial part of the T-cell pool in the tumor-rich regions, with 4.6% and 8.3% CD57+ cells among T_H_ and T_C_, respectively. The CD57+/CD57− ratio was about four-fold higher in tumor-rich vs. tumor-sparse regions (4% vs. 17%), indicating that these enriched late T-cell subsets play a role in MCL immunity. The diversity of the four investigated T-cell subsets, as measured by the Shannon Diversity Index [63], was not correlated to total CD3+ infiltration, suggesting that mechanisms driving T-cell infiltration and complexity among the T-cell subsets are distinct.

Previously, we have used an IHC analysis of bulk CD3 T cells to investigate the prognostic value of T-cell infiltration in MCL [26]. The positive association with overall survival was confirmed using an image analysis of multiplex IF in this sub-cohort, where CD3+ T cells, T_C,57−_ and T_H,57+_ were positively associated with outcome.

It was also noted that the frequency of T_C,57+_ cells correlated with increased proliferative index on the patient level. The association with Ki-67 was further confirmed in DIABLO-based data integration analysis. Here, Ki-67 expression in T_C_ was predictive of the patient having a high degree of T_C,57+_ cells. Thus, based on the association with Ki-67 in tumor cells, we speculate that T_C,57+_ cells might be associated with aggressive disease, but this is masked by the overall positive prognostic effect of the increased infiltration of total T cells. However, the investigation of smaller subsets is challenging as the variation in outcome in this population-based trial is narrow compared to clinical trial studies and thus warrants further investigations.

Independent of the accuracy of the segmentation, two-dimensional segmentation of even a thin layer of densely packed cells will inevitably contain information from adjacent T cells in direct cell-to-cell contact with the targeted cell type. Thus, in our GeoMx-based transcriptional and proteomic analysis we have enriched molecular information from our targeted cell type, but also information from the immediate cell neighborhood. These differences in molecular profiles were revealed when we explored the differences in identical T-cell subsets in different localizations of the MCL TIME, as most T-cell-related analytes will be constant, and differences will be dominated by information from adjacent cells. To explore this information on the immediate cell neighborhood, deconvolution analysis was used. Memory CD8+ T cells and CD57+ cells were more abundant among infiltrating T cells compared to adjacent T-cell-rich regions. The gene set enrichment analysis of GeoMx transcriptional data showed that the infiltrating CD8+ T cells had increased TNF signaling, consistent with a more activated phenotype. Adjacent T-cell-rich regions had higher levels of key proteins such as CD45RO and several immune suppressive factors (IDO1, VISTA, TIM-3, LAG-3 and ICOS), which may result in less T-cell expansion in regions close to the tumor.

Little is known about the molecular profile of distinct differentiation stages of T cells in cancer. With the aim to identify tentative targets on specific T-cell subsets, the molecular profiles of T_H,57−_, T_C,57−_, T_H,57+_ and T_C,57+_ were characterized among infiltrating T cells. For T_H_ cells, CTLA-4 is, as expected, a key marker which was increased on both the mRNA and protein level compared to T_C_ cells. CTLA4 is known to inhibit T-cell response by recruiting CD80 and limiting its interactions with CD28 [64]. CTLA4 is highly expressed on T_regs_ and inhibition has shown promising results in various solid cancers and in other types of lymphomas but with few clinical investigations in MCL [65]. Markers associated with T_regs_ such as FOXP3 and CD25 were, as expected, highly associated with T_H_ cells, but also more abundant in early (CD57−) compared to differentiated (CD57+) T cells.

The CD57+ T cells were associated with an expression of inhibitory molecules such as PD-L1, PD-L2 and LAG3. Recent investigations have shown weak surface expression of PD-1 and its ligands in MCL cohorts, which is potentially the reason for poor response to therapy [66,67]. Also, ICOS and 4–1BB were associated with CD57+ cells. ICOS has dual functions, as it can enhance the activity of T_regs_ while agonistic anti-4–1BB leads to the expansion of CD8+ T cells [68]. The concept of polyfunctional cells has previously been introduced by Ramello et al., they showed that KLRG1+ CD57+ senescent CD4+ T cells were expanded in breast cancer patients [42].

Although it is known that T-cell infiltration has a prognostic effect, the underlying biology associated with high and low infiltration is unknown. To explore this, DIABLO-based data integration was used to understand the differences in patients with either a high or low infiltration of total CD3. The grouping of patients was made using a cut-of defined by maximally rank statistics that results in two groups with either high CD3 infiltration and prolonged survival, or low CD3 infiltration and poor survival. The combined molecular profiles in tumor cells and T-cell subsets that assign patients to either of the two groups were defined. We show that tumor cells have an increased expression of several markers (*CD47*, *IL7R*, *NKG7*, *CD80*, *CD84*, *CSF1R*, *NLRC5/CITA*, *CD2* and *IRF3*) that may be involved in regulating the level of immune infiltration and the effect on tumor immunity. Of major interest, the clinically interesting CD47 was one of the historically first identified “don’t eat me” signals. It exerts the effect through interaction with the signal receptor protein-alpha (SIRPa), expressed on phagocytic cells, and inhibits cellular phagocytosis. The combined effect of Rituximab and CD47 blockade is currently being clinically assessed in MCL [69,70]. Markers associated with low CD3 infiltration were dominated by molecules associated with the proliferation or promotion of the aggressiveness of the disease, consistent with the poor outcome of that sub-group of patients.

Among the T_H_ and T_C_ cells, IDO1 was associated with high CD3 infiltration/favorable outcome, while GITR was associated with low CD3 infiltration/unfavorable outcome. IDO1 upregulation leads to the depletion of the amino acid tryptophan’s catabolites, which induce T-cell anergy and the expansion of T_regs_. Increased IDO1 thus enables immune suppression and promotes tumor survival [71]. IDO1 inhibitors either alone or in combination with other checkpoint inhibitors have been proposed to be potential targets in DLBCL but have, so far, not been explored in MCL [72,73,74,75]. GITR is a marker for activated CD8+ T cells [76]. The increased expression of GITR can be promoted by antigen presenting cells, through interaction with GITR-L, which further promotes T-cell survival and enhances T-cell proliferation and effector functions [77]. In our analysis, we hypothesize that the expression of GITR indicates a more active T-cell status as indicated by the simultaneous expression of GZMA and GZMB on T_C_ cells and the co-expression with BCL-XL on T_H_ cells. It has been reported that the engagement of a GITR agonist drives the production of interleukin-2 (IL-2), and pro-inflammatory cytokine interferon-γ (IFN-γ), which promotes T-cell expansion as well Treg formation [76,78,79,80,81]. IDO1 and GITR agonists/inhibitors are currently being investigated as potential targets for immunotherapy in several cancer subtypes [82,83]. Combination therapies with GITR agonists and IDO1 inhibitors are of interest and a phase-I trial for glioblastoma is currently under progress [82]. Thus, we suggest that GITR is an interesting target for patients with low T-cell infiltration, while the IDO1-STING axis may be more active in patients with high total T-cell infiltration.

As discussed above, clinicopathological data showed that T_C,57+_ was associated with more proliferative disease, and we explored the molecular mechanisms behind it. Using DIABLO, we showed that patients with high T_C,57+_ infiltration have an increased expression of *CXCL9* in their tumor cells. This was not identified in patients with a high infiltration of total T cells; thus, it may be related to the specific infiltration of T_C,57+_. *CXCL9* is a well-known chemoattractant that governs the degree of T-cell infiltration and has been shown to drive interaction between type I dendritic cells and T cells to restrict CD8+ antitumor immunity [84].

## 5. Conclusions

In conclusion, we demonstrate the power of combined deep learning-based image analysis and spatially guided omics analysis to reveal connections between the cellular composition of the MCL TIME and distinct molecular analytes in tumor and infiltrating T cells. We suggest that targeting CD47, CTLA-4 and IDO1 could be an efficient strategy to boost T-cell response in patients with high T-cell infiltration, but that specific targeting of GITR, TIGIT, LAG3, PD-L1 and PD-L2 might be more effective for patients with overall low T-cell infiltration but the presence of CD57+ T cells. We further suggest that CD57+ T-cell infiltration is associated with *CXCL9* expression in tumor cells, and specifically in high-risk patients, which needs to be investigated functionally in future studies.

## Figures and Tables

**Figure 1 cancers-16-02289-f001:**
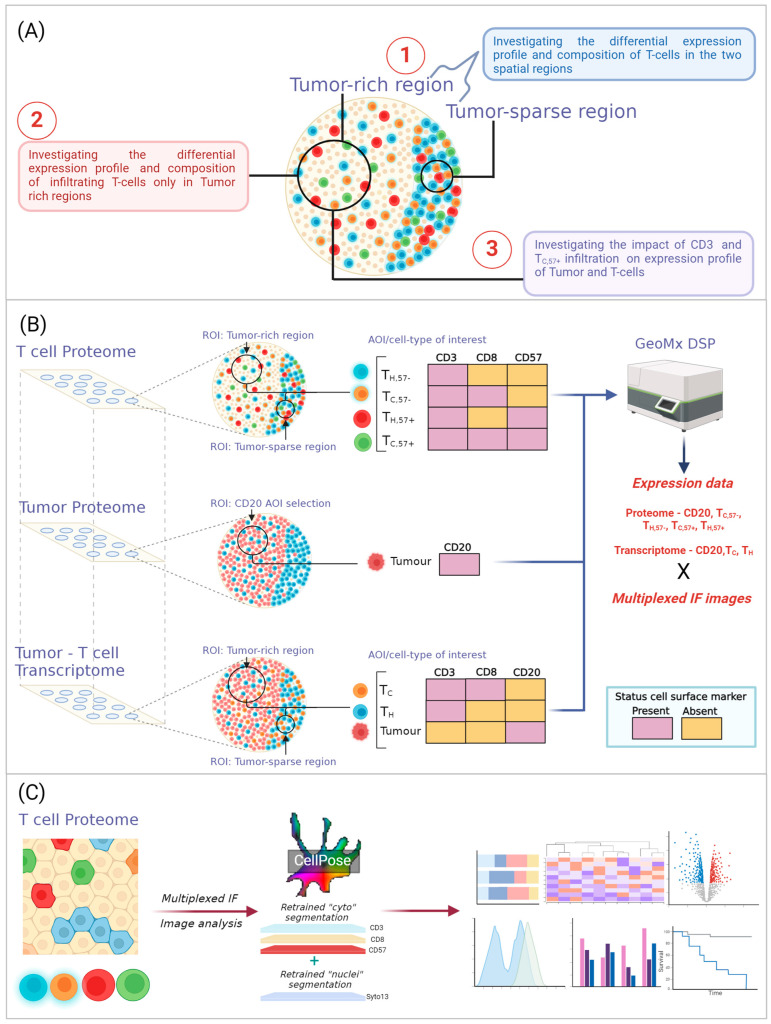
Overview of the study: (**A**) Three biological questions are explored in the study. These include (1) comparison between the differences in T-cell abundance and molecular profiles in tumor-rich and tumor-sparse regions, (2) the identification of unique mRNA and proteins on early- and late-stage T-cell differentiation stages and (3) the association between cell frequencies and molecular profiles of tumor and T cells. (**B**) Proteomic and transcriptomic data were collected from phenotypically identified cell-specific AOIs in MCL tissue from replicate TMA sections. Data collection focused on T-cell subsets using phenotypic staining of CD3, CD8 and CD57, allowing T_C,57−_ (CD57− CD3+ CD8+ T cytotoxic cells), T_H,57−_ (CD57− CD3+ CD8− T helper cells), T_C,57+_ (CD57+ CD3+ CD8+ T cytotoxic cells) and T_H,57+_ (CD57+ CD3+ CD8− T helper cells) to be enriched and collected in separate AOIs. Proteomic data collection focused on CD20+ MCL cells was achieved by staining for CD20 and CD3 to allow the identification of the separate tumor-rich and tumor-sparse compartments, where the latter is often rich in T cells. Transcriptional profiling included CD20+ MCL cells, T_H_ and T_C_ cells. (**C**) Image analysis was performed on mIF images stained for Syto13, CD3, CD8 and CD57. Cellpose models for nuclei and cell segmentation were finetuned using Syto13 staining and cell membrane markers (CD3, CD8 and CD57), respectively, which generated four cell masks. Cells were classified into four cell types by overlapping the generated cell segmentation masks with the centroid of the nuclei mask. Image-derived cell metrics were extracted and used in conjunction with expression data. https://Biorender.com was used to create the illustrations.

**Figure 2 cancers-16-02289-f002:**
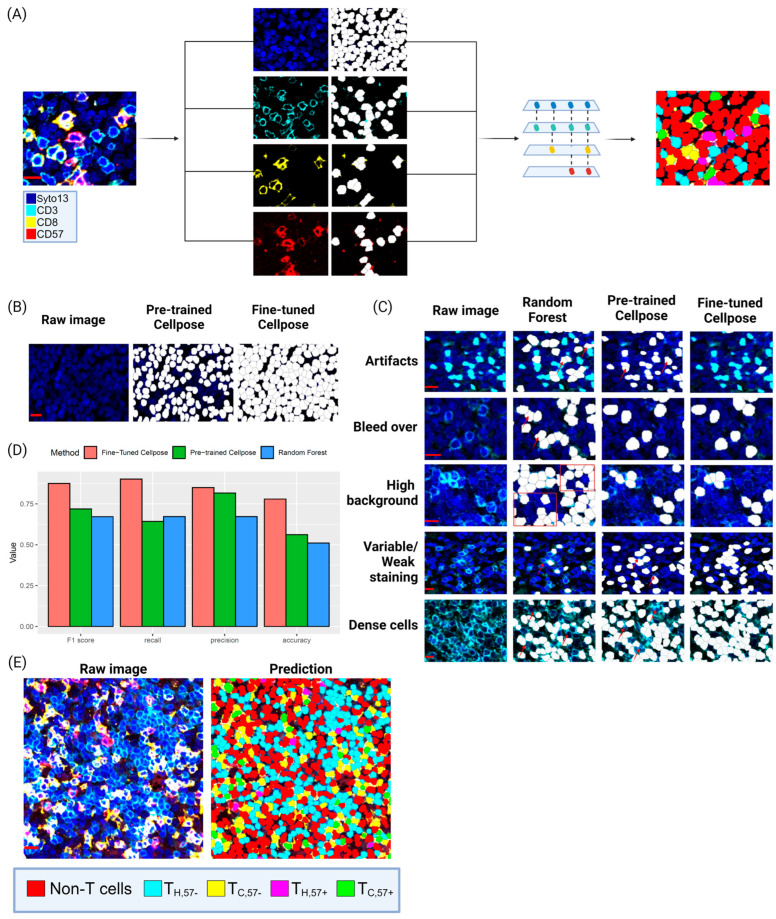
A retrained image segmentation/classification pipeline was used to classify cells. (**A**) Composite mIF image and the four channels separated with predicted segmentation masks. The segmentation masks generated from the fine-tuned Cellpose models were overlapped and projected to classify individual cells based on appearance of an individual marker. (**B**) Pre-trained vs. fine-tuned Cellpose nuclei model showing the output of the nuclei segmentation performance when the model weights are updated based on project-specific images. (**C**) Depiction of the improved performance in cell segmentation and classification of fine-tuned Cellpose cyto models (Panel 4) in comparison to pre-trained Cellpose models (Panel 3). A secondary model was developed using the masks from fine-tuned Cellpose nuclei model which was expanded to the cell boundary and a random forest classifier was trained to distribute the cells into the T-cell subtypes (Panel 2). (**D**) Quantitative comparison of F1 score, recall (sensitivity), precision and accuracy of the three models applied to a subset (*n* = 11) of cropped images. (**E**) Example of final segmentation and classification of cells into the four T-cell subtypes based on the defined workflow. Scalebar is 10 µm for all sub-figures but (**E**) that is 20 µm.

**Figure 3 cancers-16-02289-f003:**
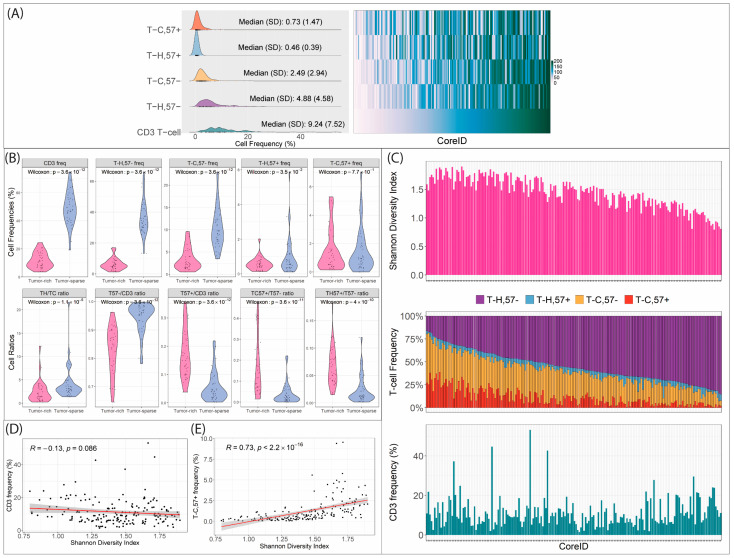
Tumor-infiltrating CD57+ T cells of both T_C_ and T_H_ subsets are present in MCL. (**A**) Histograms of tumor-infiltrating T-cell frequencies showing that tumor infiltration (%) is mainly composed of T_H,57−_ cells followed by T_C,57−_ cells. CD57+ cells were found in both the T_H_ and T_C_ compartment but were more common among T_C_. In the heatmap (right panel), cell frequencies were ranked (1–186) and sorted based on CD3+ cell frequency, to highlight relative differences. The columns represent the core-IDs. The ranked heatmap emphasizes the dependencies between total CD3 frequency and the CD57− T-cell subsets. (**B**) Paired analysis (*n* = 39) investigating the differences of T-cell frequencies in tumor-rich and tumor-sparse regions showing high variation of the CD57− T-cell subsets, with more such T cells in the tumor-sparse region. CD57+ T cells were equally abundant in the two regions. Analysis of the relative proportion compared to T57− subtypes shows that both T_C,57+_ and T_H,57+_ had a higher proportion in the tumor-rich compared to tumor-sparse area. (**C**) Shannon Diversity Index (SDI) is plotted (upper panel) in relation to the distribution of the four investigated T-cell subsets (middle panel) in tumor-rich regions. No correlation between SDI and total CD3 frequency (lower panel) is observed (also see (**D**)). The higher SDI scores are associated with presence of CD57+ subsets, particularly the T_C,57+_ (also see (**E**)), and larger variation in relative abundance of the four T-cell subsets. Lower scores were associated with dominance of mostly T_H,57−_ cells. (**D**) Spearman correlation for SDI vs. CD3 frequency and (**E**) SDI vs. T_C,57+_ cell frequency, showing that the score is positively (R = 0.73) associated with increasing CD57+ T_C_ cells. The other subtypes exhibited less pronounced correlation: T_C,57−_ (R = 0.21, *p* = 0.0036), T_H,57+_ (R = 0.23, *p* = 0.0015) and T_H,57−_ (R = −0.45, *p* < 0.00001).

**Figure 4 cancers-16-02289-f004:**
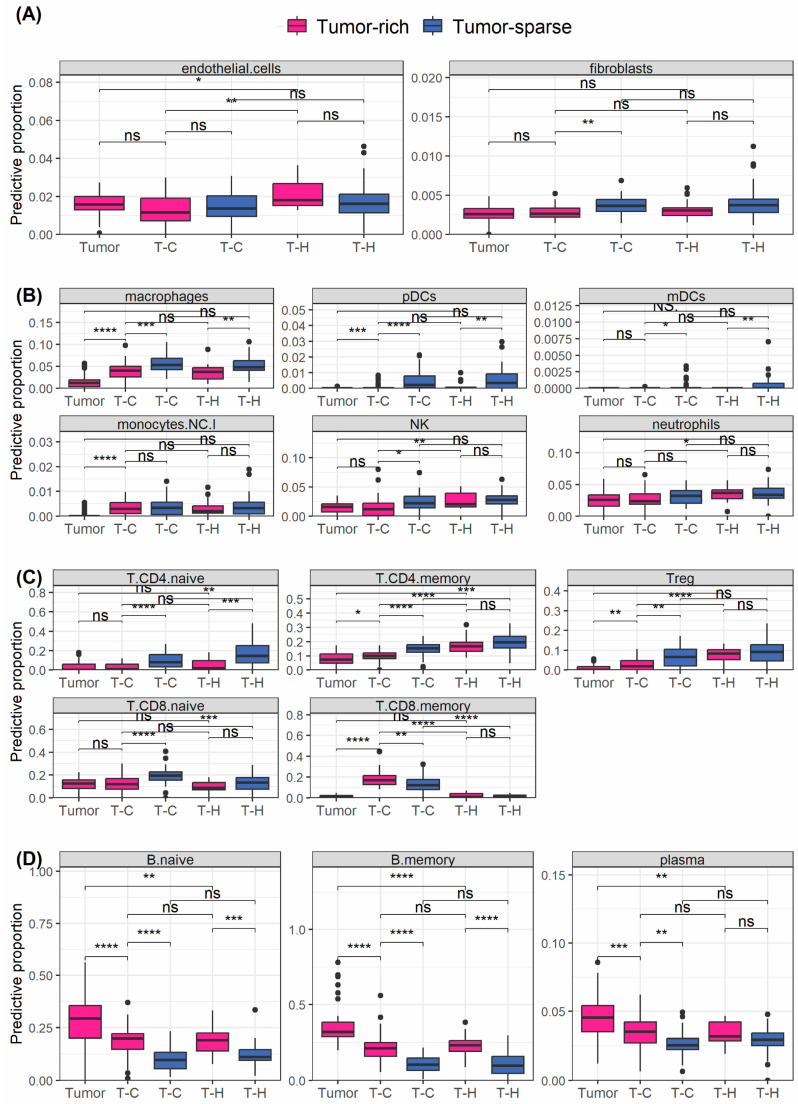
Deconvolution analysis investigating the predicted cell subtypes from tumor and T-cell AOIs. Boxplots show pairwise Wilcoxon analysis of cell types identified by deconvolution on paired (*n* = 25) transcriptome data. Data from tumor cells, and T_C_ and T_H_ in both tumor-rich and tumor-sparse regions were included in the analysis. Pink boxes indicate data sampled in tumor-rich regions, and blue boxes indicate data sampled in tumor-sparse regions. (**A**) Endothelial cells and fibroblasts, (**B**) macrophages, mDCs, monocytes (NCI, non-classical), neutrophils, NK and pDCs, (**C**) CD4+ memory T cells, CD4+ naïve T cells, CD8+ memory T cells, CD8+ naïve T cells, regulatory T cells, (**D**) naïve B cells, memory B cells and plasma B cells. ns, non-statistical significance, * *p* < 0.05, ** *p* < 0.01, *** *p* < 0.001, **** *p* < 0.0001.

**Figure 5 cancers-16-02289-f005:**
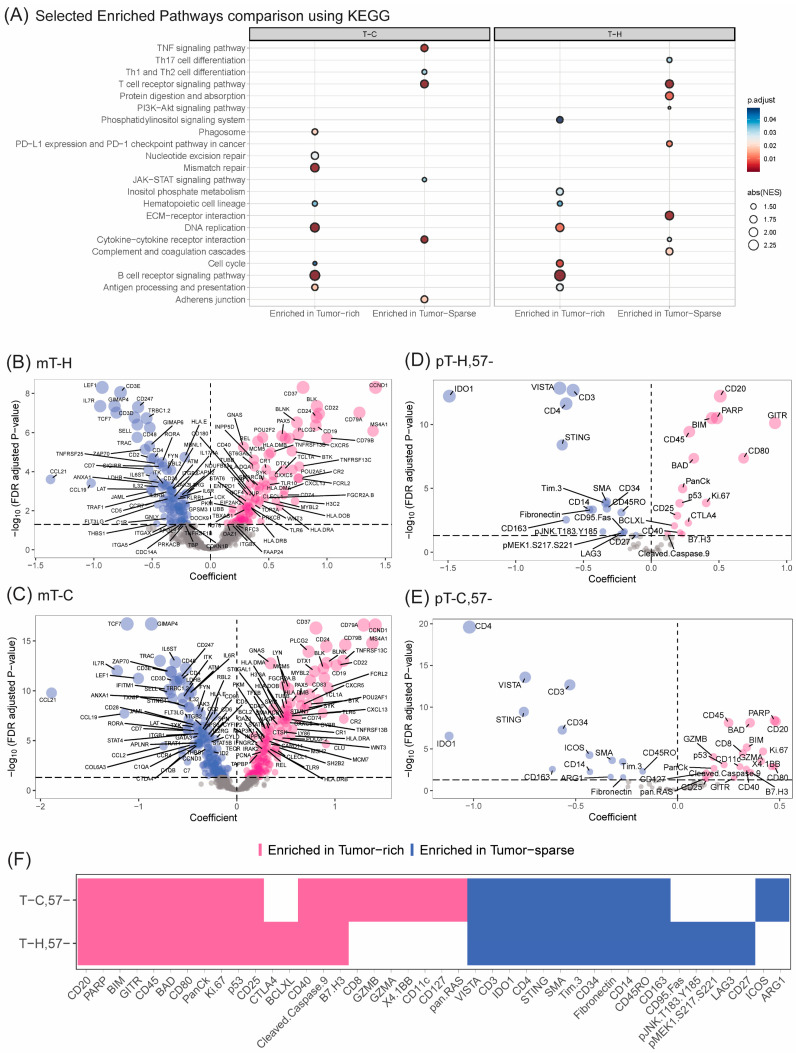
Functional differences in MCL-infiltrating T-cytotoxic and T-helper cell populations compared to adjacent T-cell-rich regions. Transcriptional data from *n* = 25 patients and proteomic data from *n* = 39 patients were used as these patients had data collected in both tumor-rich and tumor-sparse regions. (**A**) Gene set enrichment analysis of 1482 mRNA transcripts, performed for T_H_ and T_C_ cells separately. The plot has been adjusted to show pathways of interest in immuno-oncology. Paired linear mixed model (LMM) analysis (Patient ID as a random effect) comparing the differential expression of (**B**) mRNA transcripts (m) in T_H_, (**C**) mRNA transcripts (m) in T_C_ and (**D**) proteins (p) inT_H,57−_, and (**E**) proteins (p) in T_C,57−_ cells in tumor-sparse and tumor-rich regions, are visualized. (**F**) Tile plot summarizing the overlap of differentially expressed proteins in T_H,57−_ and T_C,57−_ cells in relation to spatial localization, as identified by paired LMM analysis (panel **D**,**E**). The values represent the direction of enrichment in relation to the spatial compartment. Proteins with higher abundance in the tumor-rich area are shown in pink while proteins with lower abundance are shown in blue.

**Figure 6 cancers-16-02289-f006:**
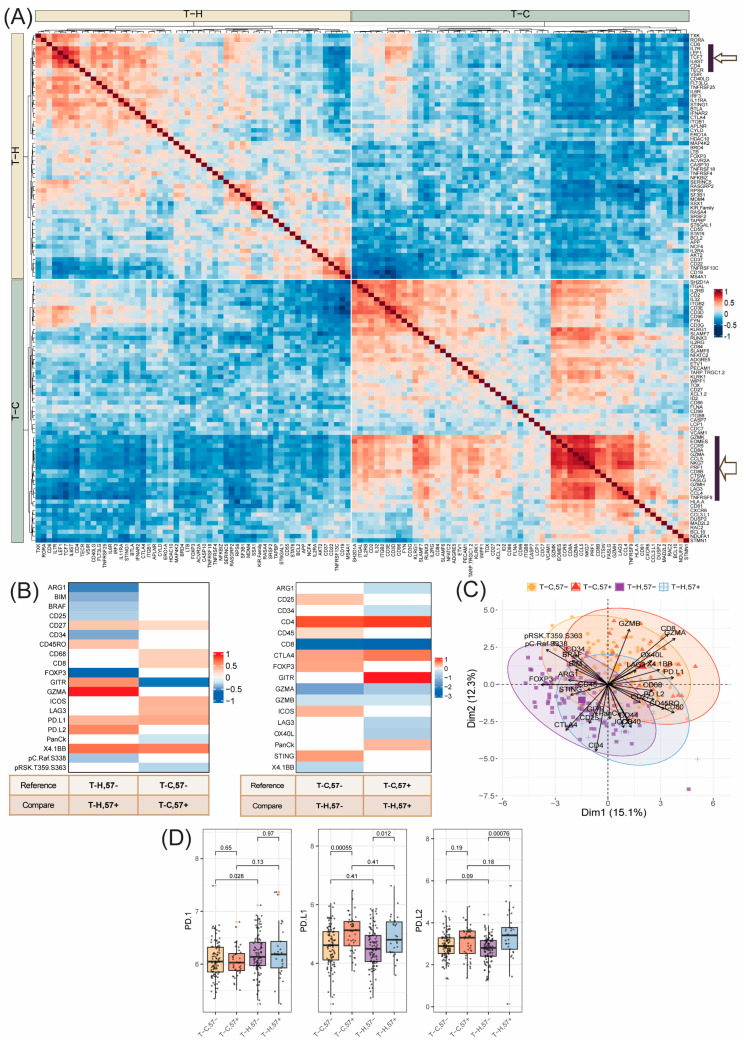
Comparison of the functional and phenotypic variation in infiltrating T-cell subtypes. (**A**) Spearman correlation plot exploring co-regulation between differentially expressed transcripts, as identified by differential gene expression using LMM analysis comparing the infiltrating T_H_ and T_C_ cells (*n* = 63). Color legend indicates positive (red) or negative (blue) correlation value. (**B**) Tile plot highlighting the differentially expressed proteins between the infiltrating four T-cell subsets (*n* = 102), as identified by ANOVA followed by Tukey-HSD test (q-value cutoff: 0.05). The values show the difference in the mean value between groups. The reference and comparison group are given in the tables below. Color code indicates relative higher (red) or lower (blue) abundance (difference in group means). (**C**) PCA biplot using the analytes identified in (**B**) showing the group segregation of the four T-cells subsets based on the magnitude and direction of differential protein expression between the two components. (**D**) Boxplot and Wilcoxon *p*-value analysis of PD-L1, PD-L2 and PD-1 expression among the four infiltrating T-cell subsets. Tumor-rich associated mean value expressions were aggregated by patient ID.

**Figure 7 cancers-16-02289-f007:**
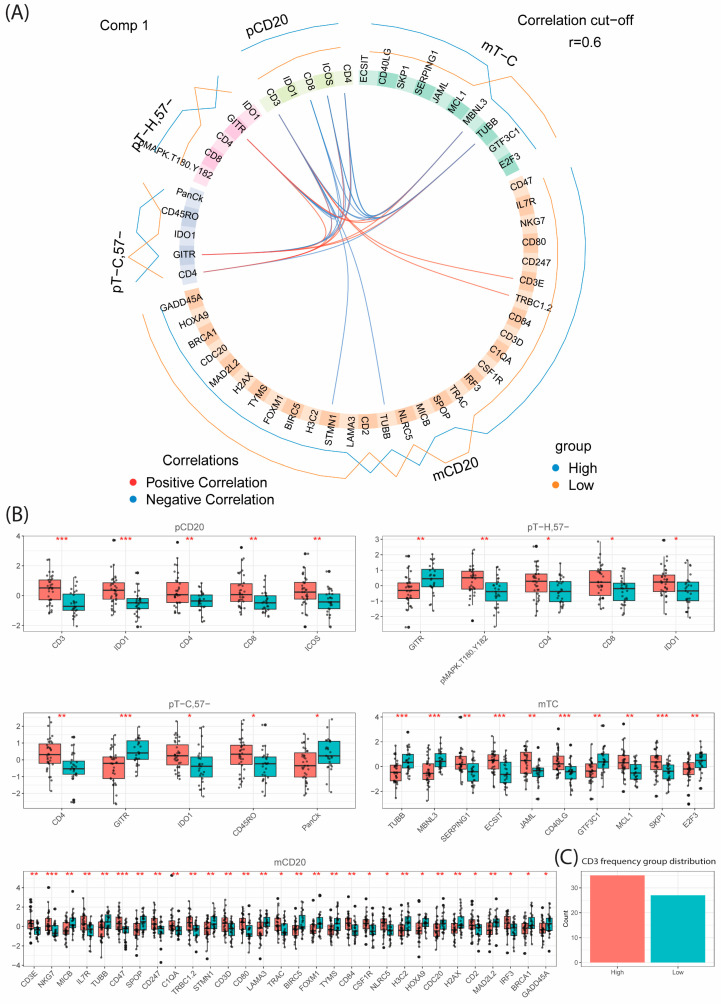
Multi-omics investigation of TIME with respect to infiltrating CD3 T-cell frequency using DIABLO. DIABLO prefers complete datasets, and to include relevant number of patients only five out of eight collected omics datasets were used resulting in data from *n* = 62 patients. Proteomic datasets included CD20 (pCD20), T_H_ (pT_H_) and T_C_ (pT_C_). Transcriptomic data included CD20 (mCD20) and T_C_ (mT_C_). (**A**) Circos plot highlighting the identified analytes in each omics dataset for the optimally selected first component. The outer blue and red lines indicate the direction of the association between the individual parameter (gene or transcript) and the CD3 frequency group (high or low, cut-of 8.4%). The inner lines connect parameters with positive (red) or negative (blue) association-based correlation analogues to Pearson (R > ±0.6). The groups were determined based on optimal cut-off for high/low CD3 T-cell infiltration based on survival analysis in Appendix A. (**B**) Boxplot analysis with *t*-test significance per omics dataset of the analytes identified in (**A**). * *p* < 0.05, ** *p* < 0.01, *** *p* < 0.001. (**C**) Bar plot distributions of the high (*n* = 35) and low (*n* = 27) infiltration groups used for this analysis.

**Figure 8 cancers-16-02289-f008:**
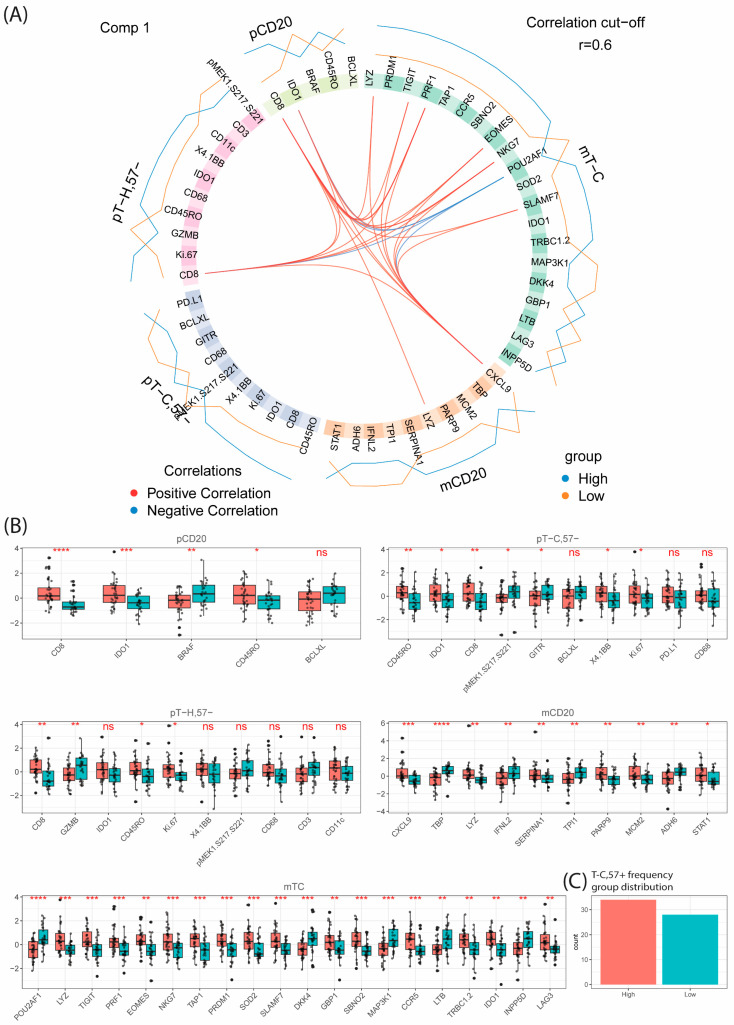
Multi-omics investigation of TIME with respect to infiltrating T_C,57+_ frequency using DIABLO. DIABLO prefers complete datasets, and to include a relevant number of patients only five out of eight collected omics datasets were used, resulting in data from *n* = 62 patients. Proteomic datasets included CD20 (pCd20), T_H_ (pT_H_) and T_C_ (pT_C_). Transcriptomic data included CD20 (mCD20) and T_C_ (mT_C_). (**A**) Circos plot highlighting the identified analytes in each type of omics data for the optimally selected first component. The outer blue and red lines indicate the direction of the association between the individual parameter (gene or transcript) and the T_C,57+_ frequency group (high or low, using median = 0.686% as cut-off). The inner lines connect parameters with positive (red) or negative (blue) association-based correlation analogues to Pearson (R > ±0.6). (**B**) Boxplot analysis with *t*-test significance per type of omics data of the analytes identified in (**A**). (**C**) Bar plot distribution of the high (*n* = 34) and low (*n* = 28) infiltration groups used for this analysis. ns, non-statistical significance, * *p* < 0.05, ** *p* < 0.01, *** *p* < 0.001, **** *p* < 0.0001.

**Figure 9 cancers-16-02289-f009:**
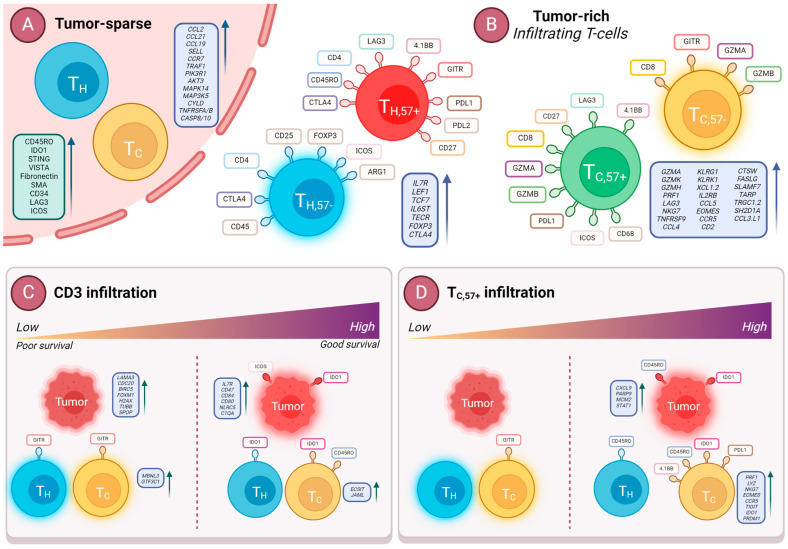
Summary of results. (**A**) Proteins (green box) and transcripts (purple box) upregulated in tumor-sparse regions (data shown over pink background) compared to tumor-rich regions. (**B**) Key proteins displayed on surface of the individual cells and mRNAs (purple boxes) on each T-cell subset. The left box refers to upregulated transcripts in T_H_ subsets and the right box refers to upregulated transcripts in T_C_ subsets. (**C**) Main proteins and mRNAs predictive of high or low CD3+ T-cell infiltration. Purple boxes indicate upregulated transcripts in each indicated cell type. (**D**) Main proteins and mRNAs predictive of high or low T_C,57+_ infiltration. Purple boxes indicate differentially upregulated mRNAs in each indicated cell type. Biorender.com was used to create the illustrations.

## Data Availability

The datasets used and/or analyzed during the current study are available from the corresponding author on reasonable request. Scripts and codes used for image analysis are available in https://github.com/Nilsson-D/Image-analysis-GeoMx-DSP (accessed on 12 June 2024).

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
