# Peer review of "Quantification and Profiling of Early and Late Differentiation Stage T Cells in Mantle Cell Lymphoma Reveals Immunotherapeutic Targets in Subsets of Patients"

_cancers, 2024, doi:10.3390/cancers16132289_

Round 1

Reviewer 1 Report

Comments and Suggestions for Authors

The authors use a variety of modern spatial profiling technologies to study the tumor immune microenvironment of mantle cell lymphoma. Lymphomas represent a fairly challenging family of cancers to study with spatial techniques due to close packing of cells. The authors attempt to connect multiple measurement modalities, and in the scope of a study that will not likely involve wet lab experiments, the authors perform a fairly extensive amount of analysis and data integration. I have moderate concerns about the improvements the authors claim and some of the conclusions they draw. If the authors are able to address some of the concerns regarding reproducibility and transparency of the methods, I believe the study is suitable for publication.

Major comments

1. The tumor rich/sparse classification is not transparent. The description provided in lines 116-119 is not sufficient for reproducibility and no data is provided on the descriptive differences between these ROIs. This should be clarified as it is the basis of the study.

2. Claims about improvement of the segmentation model. There is not enough data supporting this claim. Figure 2 is not quantitative, only a small area of images is provided. For example, in line 511, authors claim that relatively increased levels of CD68 indicate that macrophages are proximal to Tc,57+. However, if CD57 staining is poor, then CD57+ cells may systematically have worse segmentation and be more likely to capture adjacent cell signal than CD57- cells.

3. Transcriptome analysis. Based on the description in lines 370-373, the deconvolution strategy appears to rely on imperfect segmentation leading to "information also from the most adjacent cells". I have 2 issues here, first the observation of cells like macrophages and DCs being more abundant in tumor-sparse compared to tumor-rich regions may be trivial because the tumor-rich region "adjacent cells" will obviously be tumor (which is deconvolved into the B cell types). Second, was the deconvolution performed before all of the GSEA analysis as well? Otherwise you have "adjacent cell" signal polluting the T cell GeoMx data in different ways between the two regions.

4. Figure quality. The figure quality should be improved before publication. Text is difficult to read, fonts are different types and sizes, figures are spread over multiple pages and have strange borders. For example, Figure 5F should contain information on fold changes or significance, instead there is just blocks of color with no information added. The color used for tumor-rich/sparse is not consistent. Figure 6A is difficult to read. Circos plots are not explained. Figure 7 and 8 contain many comparisons that are not significant, why?

Minor comments

1. In your adjacent tissue slice analysis, how well is CD3 and CD20 signal correlated between different mIF analyses? In general, the basic characterization of the staining and imaging is not detailed. Are there batch effects between patient samples, and how significant are inter-sample vs intra-sample comparisons?

2. The figure captions have errors (i.e. Figure 4 has no section D), and the figure captions are too long. Captions should minimize repeating of information from the text.

3. Linear regression is probably not an appropriate test for Figure3E, perhaps you should compare between 4-5 strata defined by SDI.

4. Language that suggests causality cannot be supported, such as line 338: "Thus, increased T cell infiltration per see will not automatically lead to the presence 338 of both T H and T C , or well differentiated T cells."

Comments on the Quality of English Language

There are a number of minor editing errors, such as:

line 322 were->where

line 325  percentage of Tc vs Th instead of Tc among Th

line 338 see -> se

The paper should be thoroughly proofread

Reviewer 2 Report

Comments and Suggestions for Authors

The manuscript from Lokhande and co-authors is of high interest with very interesting results and exciting methodology/machine learning. It is a very dense manuscript with many figures and supplementals which were sometimes not easy to follow because not mentioned in the manuscript or not in order. To ease the reading and make it more enjoyable I will suggest to the authors to organize their figures and supplementals to have them appear in order with the results section. Otherwise, I have minor comments/questions below. 

Introduction:

- Why did you choose to dichotomize based on CD57 expression? I think it can be great to have some background behind this reasoning in the introduction if this choice was based on previous studies.

Methods:

-  Do you know why you have more male than female in the cohort? Is the cancer more prevalent in male? I mean, is your cohort reflecting the incidence of the cancer per gender?

- Because not all samples were used for all the different analysis, it can be good to indicate in the figure legend how many samples were used. 

Results:

It will help the reader to appropriately reference the numerous figures, particularly the supplemental ones. In the results, we pass from S2A to S2E (which is indicated as D as of now), and then we go back to S2B and S2C, but S2D is not referenced in the manuscript.

- Is line 291 a section title? If yes, please use italic format (it applies for all the title downstream). Also, the title indicates CD57+ being a significant proportion of infiltrated cells however, line 292 to 296, the results showed higher frequencies of the TH57- (56%) and TC57- (28%) whereas 57+ subsets were only at 4.6 and 8.3 % which does not match with the title of this section, or it mislead the reader.

-As CD57 is a known mature marker of lymphocytes, was it expected to find higher abundance of less mature lymphocytes in the tumor-rich sections? How the tumor-sparse sections look like?

- Figure S2, panel D is there twice instead of panel D and E.

- Lines 298 to 301: I don’t understand the statement that because no correlation is observed between the frequency of CD3pos cells and the T57+/T57- ratio, the infiltration of highly differentiated cells is independent of T cells infiltration. In the S2C panel, there are strong correlation between the frequency of each cell (TC57-, TH57-, TC57+, TH57+) and the frequency of CD3 cells which I believe makes sense as they are all CD3 positive cells to start with, sorry if I misunderstood. Can the authors explain a little bit more this statement?

- Figure 3A, I don’t understand the axis of the plots and the panel is not mentioned in the results section. Are the number “0”, “20”, “40” frequencies? If yes, frequencies of total lymphocytes? As it is a histogram, is the Y-axis the number of cells? Also, the ranked heat-map should be more detailed in the legend. What are the values “200, 150…” and colors representing? Is each column a participant?

- Can the panel 3B be placed under panel 3A instead of 3C? It might be because of the format but the panel 3B is hard to read because too blurry.

- Line 310: how can TC57+ can be higher among T57- knowing that they are in separated group. All the results so far showed that T57- are more abundant than TC57+, I don’t understand this statement, can the author explain it better?

- Figure 3E, did you assess as well the correlation between the SDI and the frequencies of the other T cell subsets (TC57-, TH57- and TH57+)?  

- The supplemental figure 3 is not mentioned/referenced in the main text.

- Line 346, was the Ki67 expression measured in the tumor cells, or in the tumor lymphocytes infiltrated cells? It is not clear.

- Line 357, I believe the referenced figure is not the correct one as S2D is about Ki67 and the line 357 talks about another analysis.

- Can the authors explain a little bit more the results mentioned line 363 to 367? How these results led to the section title “Spatially guided comparison of TH and TC subtypes in tumor-rich versus tumor-sparse regions of 361 MCL tissue”? Is it in reference to the supplemental figure 3?

- Figure 4, even though it is composed of 4 panels and within each panels multiple plots, only the panel 4B is mentioned in the results. Please mention/describe/interpret all the panels presented in the figure to guide the reader though the figure.

-  Figure 5, what is “mT” compared to “pT”? Is it to differentiate transcriptome to proteome? Can it be mentioned in the legend?

- Line 419, all data should be shown, it can be added into the supplemental figures.

- Line 430 to 433: be cautious with the exhaustion phenotype as it requires not only a sustained and intense expression of inhibitory receptors, but also a proof of lack of effector function (cytokines secretion, cytotoxic function, proliferative capability) as well as impaired metabolism and others. Upon activation, lymphocytes will express inhibitory markers such as LAG3, TIM3, TIGIT, CTLA4 or PD1 (alone or in combination) to allow its own immunoregulation.

- Line 534: I believe the referred figure is S5E instead of 5B, please correct.

- Line 541/542 and Figure S6: did you look at the LAG3 expression on pT-C57+ and pTH57+ in tumor sparse area? Figure 2B was showing higher frequencies of CD57 expressing TH and TC in tumor sparse area, it will be interesting so see if they have similar profile compared to the tumor-rich area.

- Line 569, is there a particular reason to use a cut-off of 8.4% to compare high vs low T-cell infiltration?

- Figure 7, are the group high and low like the tumor enriched vs tumor sparse? I did not get that part clearly. 

- Figure 9A are the reported transcripts from the T cells, the green list from the TH and the purple one the TC? It is not clear. I suggest to separate the 9A and 9B as it is for 9C and 9D as I was confused at first thinking that 9A and 9B were to be compared to each other which was not making sense.

Discussion:

For some of the analysis the number of samples was small, and you were showing high heterogeneity of the T cells subset distribution across participants. Did you check for the small number of samples analysis if the samples were coming from the same “distribution type” (figure 3C)? The DIABLO analysis was performed on high versus low CD3 infiltration, was it based on the figure 3C? I am a little confused on how it was determined. I was wondering if the authors can clarify or discuss this point, to know if the results are representative of all type of T-cell frequency based on the figure 3C.   

Round 2

Reviewer 1 Report

Comments and Suggestions for Authors

I find the revision to be mostly satisfactory, with just a few remaining comments.

I cannot assess the quality of the revised figures as they were not provided in the review. However, the range of font sizes is found within a single figure, i.e. figure 5 titles for panels B-E vs plot title in panel A, so I doubt that it is an artifact of the upload. It is not disqualifying but it is room for improvement.

Line 71-73, English is not clear. Is CD57 a marker of senescent or proliferating T-cells, or is CD57+ T cells a marker?

Line 289, the text remains a "visual comparison" so there is still a claim of superiority. Since you have presumably performed the random forest and the modified fine-tuned Cellpose, and there is still a claim of improved performance, it should be compared quantitatively in a simple supplemental figure panel.

The CD57 confusion could be solved by adding detail to the methods. My concern is that when a CD57+ cell is called, does the segmentation of these cells depend on the CD57 marker? In that case, the quality of the CD57 marker will result in a difference between the relative size of CD57+ vs CD57- T cells, i.e. if CD57 is a noisy signal, then CD57+ cells may be systematically larger than CD57- cells and may therefore automatically bleed over more signal from their neighbors. Normally I would not care about this detail but since CD57+/- is the basis of this study I am concerned for this detail. Sample images and a quantitative comparison of morphology/cell size between CD57+/- cells would fix this for me.
